# SpelsNet: Surface Primitive Elements Segmentation by B-Rep Graph Structure Supervision

**Kseniya Cherenkova**
SnT, University of Luxembourg, Artec3D
kseniya.cherenkova@uni.lu

**Elona Dupont**
SnT, University of Luxembourg
elona.dupont@uni.lu

**Anis Kacem**
SnT, University of Luxembourg
anis.kacem@uni.lu

**Gleb Gusev**
Artec3D
gleb@artec3d.com

**Djamila Aouada**
SnT, University of Luxembourg
djamila.auoada@uni.lu

## Abstract

Boundary Representation (B-Rep) is the standard approach for modeling shapes in Computer-Aided Design(CAD). We present SpelsNet, a neural architecture for segmenting 3D point clouds into surface primitive elements under topological supervision of its B-Rep graph structure. We also propose a point-to-BRep adjacency representation that allows for adapting conventional Linear Algebraic Representation of B-Rep graph structure to the point cloud domain. Thanks to this representation, SpelsNet learns from both spatial and topological domains to enable accurate and topologically consistent surface primitive element segmentation. In particular, SpelsNet is composed of two main components; (1) a supervised 3D spatial segmentation head that outputs B-Rep element types and memberships; (2) a graph-based head that leverages the proposed topological supervision. To train SpelsNet with the proposed point-to-BRep adjacency supervision, we extend two existing CAD datasets with the required annotations, and conduct a thorough experimental validation on them. The obtained results showcase the efficacy of SpelsNet and its topological supervision compared to a set of baselines and state-of-the-art approaches.

## 1 Introduction

Creating a structured and editable Computer-Aided Design (CAD) representation Mortenson [2006], Shah and Mäntylä [1995] from an unstructured 3D scan (*e.g.* point cloud) is a core challenge, often referred to as *reverse engineering*. This field has a long history of extensive research due to its numerous commercial applications Abella et al. [1994], Varady et al. [1997], Bénière et al. [2013], Liu et al. [2023]. Modern CAD workflows commonly use *Boundary Representation* as the primary format to model complex shapes Lambourne et al. [2021], Guo et al. [2022]. The wide adoption of B-Rep in most CAD software and recent advances in neural point cloud representations challenges the reverse engineering research towards the problem of learnable B-Rep inference from point clouds Liu et al. [2023], Guo et al. [2022], Yan et al. [2021], Huang et al. [2021].

Boundary Representation (B-Rep) is a collection of connected surface elements with their geometric definitions in a form of parametric surfaces, curves and points Shah and Mäntylä [1995]. The topology of these elements is also described by the connection of *face*, *edge* and *vertex* components. Thus, face is a bounded surface, edge is a bounded curve, and vertex is a realisation of a 3D point DiCarlo et al. [2014]. B-Rep is a compact representation and it retains more structural information about an object than a point cloud. Most popular approaches to reverse engineer B-Reps from point clouds follow a *segmentation-fitting* paradigm, *i.e.* the point cloud is firstly segmented into surface patches,

38th Conference on Neural Information Processing Systems (NeurIPS 2024).

and then parameterized by fitting a specific surface type Sharma et al. [2020], Li et al. [2019], Yan et al. [2021], Huang et al. [2021]. However, existing segmentation-based approaches mostly deal with either surface patches or boundary curves, ignoring the full B-Rep structure. This often leads to inaccurate and disjoint reconstruction of its elements Sharma et al. [2020]. To address this, ComplexGen Guo et al. [2022] modeled the B-Rep as a *chain complex* Hatcher [2002] and formulated the prediction of validness and primitive types as classification tasks to recover corners, curves, and patches together with their mutual topological features. Assembled in a probabilistic graph constraints, this topological information is further used in a time consuming post-processing topological and geometrical optimization to recover plausible geometry.

In this work, we propose to exploit the topological information from B-Rep as a direct neural supervision within a Graph Neural Network (GNN) paradigm. To incorporate this supervision together with geometric data into a single learnable pipeline, we consider the Linear Algebraic Representation (LAR) of B-Rep chain complex DiCarlo et al. [2014]. LAR is defined on B-Reps and fully encodes their chain complex in sparse and compact matrices offering desirable learning properties. To enable direct supervision on point clouds using LAR, we adapt it to the point cloud domain and propose a novel *point-to-BRep adjacency* representation. Furthermore, we design a novel end-to-end trainable network architecture, named SpelsNet, for the inference of B-Rep elements from point clouds. SpelsNet is composed of a spatial and a topological component leveraging both the classical segmentation and the proposed point-to-BRep adjacency supervision signals, respectively. The contributions of this work can be summarized to:

- A novel LAR-based representation of B-Rep chain complex adapted to point clouds, called point-to-BRep adjacency, that allows for direct neural supervision of B-Rep topological information on point clouds. To the best of our knowledge, we are the first to propose a direct B-Rep chain complex supervision on point clouds;
- SpelsNet, an end-to-end trainable architecture for B-Rep element segmentation from point clouds. SpelsNet unifies 3D spatial and graph neural networks in a single design and exploits the proposed LAR-based point-to-BRep adjacency supervision in addition to the classical B-Rep element segmentation supervision;
- Extended versions of two existing CAD datasets ABCParts Li et al. [2019] and CC3D Cherenkova et al. [2020]. The new versions, called ABC-VEF and CC3D-VEF, include the proposed LAR-based point-to-BRep adjacency representation on the point clouds and will be made publicly available to enable further research;
- A thorough experimental validation showcasing the superiority of the proposed method over multiple baselines and state-of-the-art approaches.

The rest of the paper is organized as follows: Section 2 discusses the related works. In Section 3, we provide background on B-Rep chain complex and present the proposed point-to-BRep adjacency representation. Section 4 offers a detailed description of the proposed SpelsNet network. The experiments are reported and discussed in Section 5. Finally, Section 6 concludes the paper and provides perspectives for future works.

## 2   Related Works

Existing research on Scan-to-Brep often focuses on specific aspects of the problem, such as enhancing segmentation, improving surface fitting, or refining topology. We categorize these approaches accordingly in the following discussion.

Starting with Efficient Ransac Schnabel et al. [2007], which progressively estimates primitive parameters within point cloud in a sample consensus paradigm, continuing with data-driven learning methods Li et al. [2019], Sharma et al. [2020], that train point-based neural networks to assign patch primitive types and parameters to each input point, it became common to solve Scan-to-Brep problem in two-phase manner, namely, decomposition (segmentation) and fitting. PrimitiveNet Huang et al. [2021] proposes to treat primitive types as semantic classes and use adversarial learning to guide feature enrichment for better surface property representation. HPNet Yan et al. [2021] runs a mean-shift clustering over the hybrid representations that are combined by learnt weights.

Compared to SPFN Li et al. [2019], ParSeNet Sharma et al. [2020] constructs an additional SplineNet component to extend the set of supported surface primitives with bspline surfaces. BPNet Fu et al.

[2023] discards the primitive types and approximates all surface patches with bspline surfaces. QuadricsNet Wu et al. [2023] defines a fitting process in a form of quadrics. Several approaches focus solely on edge reconstruction to generate wireframes. For instance, NerVE Zhu et al. [2023] utilizes a neural volumetric edge representation for piecewise linear curves extraction, while DEF Matveev et al. [2022] regresses a continuous distance field to the closest edge supplemented by spline-based curve extraction. SepicNet Cherenkova et al. [2023] builds an end-to-end trainable network, where the curve fitting is formulated in a primitive-differentiable manner. Mentioned aboveFu et al. [2023], Wu et al. [2023], Zhu et al. [2023] can be considered as alternative representations, though interesting, but fall off the traditional B-Rep structure.

A major challenge in previous work has been the discontinuity of predicted surface elements, often requiring extra post-processing. While ParSeNet Sharma et al. [2020] offers an optional refinement module, Li et al. [2023] tackle this issue directly by simultaneously detecting surfaces and edges using a two-branch network. AutoGPart Liu et al. [2022] presents a generalizable approach for 3D part segmentation with geometric priors, potentially improving continuity.

We propose to leverage B-Rep inherent graph structure, with nodes representing elements like vertices, edges, and faces, directly within a Graph Neural Network by developing a unified representation for both spatial and graph domains. This allows us to directly incorporate B-Rep topological relationships and node features, such as element type and potentially even edge classifications (e.g., sharp vs. smooth).

Various graph-based learning techniques, particularly Graph Convolutional Networks (GCNs) and Message Passing Neural Networks (MPNNs), have been applied to diverse data types such as part assemblies, social networks, etc Zhou et al. [2020]. Notably, GCNs have been successfully employed in tasks like automatic mating prediction in CAD assemblies Jones et al. [2021] while MPNNs have been adapted for specific B-Rep tasks, such as face segmentation in BrepNet Lambourne et al. [2021]. While Spectral Convolutional Networks offer potential *e.g.*, Smirnov and Solomon [2021], their computational cost can be prohibitive for large graphs. ComplexGen Guo et al. [2022] predicts validity and primitive types while recovering topological features, but relies on time-consuming post-processing.

Our approach employs a unified segmentation framework with trainable LAR characteristic matrices to directly learn the B-Rep structure.

## 3    Point-to-BRep Adjacency Formulation

Given an input 3D point cloud, our method aims to identify and extract individual elements of the corresponding CAD model's Boundary-Representation (B-Rep), *i.e.* vertices, edges, and faces. In addition, the method determines the primitive type of these elements and their topological connectivity information. We present the essential background on B-Reps, including theis key elements and their connectivity relationships, then detail our approach for adapting them to point cloud data.

### 3.1    Background on Boundary-Representation (B-Rep) Chain Complex

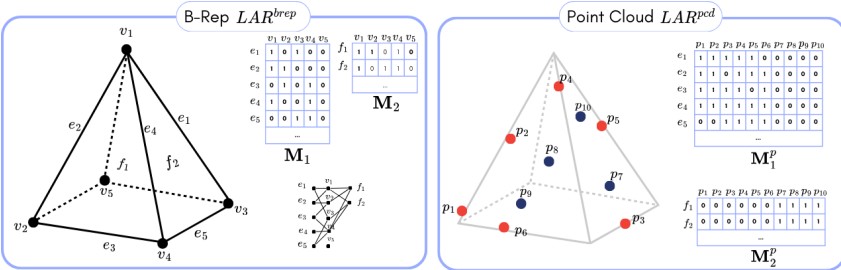

Figure 1: *Left panel:* The B-Rep elements and their topological connectivity in the form of LAR$^{brep}$ with edge-vertex characteristic matrix $\mathbf{M}_1$ and face-vertex characteristic matrix $\mathbf{M}_2$. *Right panel:* The proposed point-to-BRep adjacency representation LAR$^{pcd}$ and its characteristic matrices $\mathbf{M}_1^p$ and $\mathbf{M}_2^p$ for points on edges (in red) and faces (in blue).

A B-Rep $\mathcal{B}$ is composed of three elements, namely faces, edges and vertices. $V = \{v_i\}$ is as the set of all vertices where $N_v$ is the number of vertices. Similarly, $E = \{e_i\}$ is the set of $N_e$ edges, and $F = \{f_i\}$ the set of $N_f$ faces. Each edge $e_i$ is defined by a curve of a specific type (*e.g.* line, arc, etc). Each face $f_i$ is also defined by a surface (*e.g.* planar, spherical, etc) and its corresponding parametric description. Crucially, a B-Rep $\mathcal{B}$ not only stores information about these individual elements but also encapsulates the connectivity information between vertices, edges, and faces. This connectivity information is essential for defining the overall topology of the model. As shown in an example in the left panel of Figure 1, a pyramid's B-Rep data structure stores information about its vertices, edges, and faces. The topological relationships between these elements are represented in a Vertex-Edge-Face graph, defining the pyramid's overall topology.

Formally, the B-Rep can be described as a chain complex $\mathcal{C} = (V, E, F, \delta, \Pi)$ of order $d = 3$, where boundary operator $\delta$ connects the elements of different orders, and $\Pi$ is a set of possible attributes (refer to Hatcher [2002] for more details). For instance, $\delta_2 f_i \in E$ gives the edges which define the boundary of a face $f_i$ and $\delta_1 e_i \in V$ the end vertices of the edge $e_i$. In other words, each element set $V, E, F$ induces a corresponding vector space $\mathbb{V}, \mathbb{E}, \mathbb{F}$ and its boundary transition $\mathbb{F} \xrightarrow{\delta_2} \mathbb{E} \xrightarrow{\delta_1} \mathbb{V}$. Further, we use of the Linear Algebraic Representation (LAR) described in DiCarlo et al. [2014], a convenient and efficient representation that supports topological constructions that typically arise in a cellular decomposition of B-Rep space. Formally, LAR encodes a chain complex $\mathcal{C}$ of order $d$ by a set of binary characteristic matrices $\mathbf{M}_u$, with $1 \leq u < d$, encoding the incidence of B-Rep elements. These matrices provide a convenient and sparse-compact form for defining topological relations of B-Rep elements. For a B-Rep chain complex (*i.e.* of order 3) there exist two characteristic matrices, $\mathbf{M}_1 = \mathbf{\Delta}(E, V) \in \{\mathbf{0}, \mathbf{1}\}^{N_e \times N_v}$ and $\mathbf{M}_2 = \mathbf{\Delta}(F, V) \in \{\mathbf{0}, \mathbf{1}\}^{N_f \times N_v}$. Here, $\mathbf{\Delta}(E, V)$ assigns 1 to $\mathbf{M}_1[i, j]$ if an edge $e_i \in E$ is bounded by vertex $v_j \in V$ and 0, otherwise. Similarly, $\mathbf{\Delta}(F, V)$ operates on faces and vertices to construct $\mathbf{M}_2$. An example of a characteristic matrix $\mathbf{M}_1$ can be found in the left panel of Figure 1. As mentioned in DiCarlo et al. [2014], $\mathbf{M}_1$ and $\mathbf{M}_2$ can fully characterize the B-Rep chain complex and can be used to obtain the following incidence and adjacency matrices,

$$\mathbf{A}_{ff} = \mathbf{M}_2 \mathbf{M}_2^T; \quad \mathbf{A}_{ee} = \mathbf{M}_1 \mathbf{M}_1^T; \quad \mathbf{A}_{vv} = \mathbf{M}_1^T \mathbf{M}_1 . \tag{1}$$

Here, $\mathbf{A}_{ff}$ represents the adjacency of faces in a B-Rep, that is the faces that are bounded by a common edge. Similarly, $\mathbf{A}_{ee}$ provides the edges that are bounded by a common vertex and $\mathbf{A}_{vv}$ provides the vertices that bound a common edge. Note that as explained in DiCarlo et al. [2014], the characteristic matrices are typically sparse for actual B-Rep chain complexes, so they can be stored and operated in memory-efficient Compressed Sparse Row (CSR) format. The product and transposition of such CSR matrices, needed to compute the boundary, adjacency and incidence operators between such linear spaces, are intrinsically efficient, since the sparse matrix-vector (SpMV) multiplication is linear in the size of the output.

### 3.2 Proposed Point-to-BRep Adjacency Representation

The LAR representation is the core concept for our proposed topological supervision. As LAR are defined on B-Rep, we formulate a mechanism to transfer LAR elements from B-Rep to point cloud domain such that the learning of the B-Rep characteristics from a point cloud can be facilitated. We use the terminology LAR$^{brep}$ and LAR$^{pcd}$ to distinguish between the LAR of B-Rep and its point cloud reformulation.

The right panel of Figure 1 depicts an example of the topological transfer to a point cloud. Let $\mathbf{P} = \{p_i \in \mathbb{R}^{d_p} | i = 1..N_p\}$ be a point cloud composed of $N_p$ points, where $d_p$ denotes the dimension of point features. We define the characteristic matrix $\mathbf{M}_1^p = \mathbf{\Delta}_p(E, \mathbf{P}) \in \{\mathbf{0}, \mathbf{1}\}^{N_e \times N_p}$ in LAR$^{pcd}$ as a binary matrix with rows representing the edges of the B-Rep and columns the points of $\mathbf{P}$. Here, $\mathbf{\Delta}_p(E, \mathbf{P})$ assigns the value of $\mathbf{M}_1^p[i, k]$ to 1 if a given point $p_i \in \mathbf{P}$ belongs to an edge $e_k \in E$. This function also sets the value $\mathbf{M}_1^p[i, l]$ to 1 if the edge $e_l \in E$ is adjacent to $e_k$. Otherwise, the value is set 0. Similarly, the characteristic matrix $\mathbf{M}_2^p = \mathbf{\Delta}_p(F, \mathbf{P}) \in \{\mathbf{0}, \mathbf{1}\}^{N_f \times N_p}$ in LAR$^{pcd}$ is also a binary matrix. The rows correspond to the faces of the B-Rep and the columns to the points of the point cloud. Here also, $\mathbf{\Delta}_p(F, \mathbf{P})$ operates in the same way as $\mathbf{\Delta}_p(E, \mathbf{P})$ but on faces instead of edges. Note that the characteristic matrices in LAR$^{pcd}$ encode the per-point B-Rep edge and face memberships along with their connectivity that are essential elements of the B-Rep structure. As in

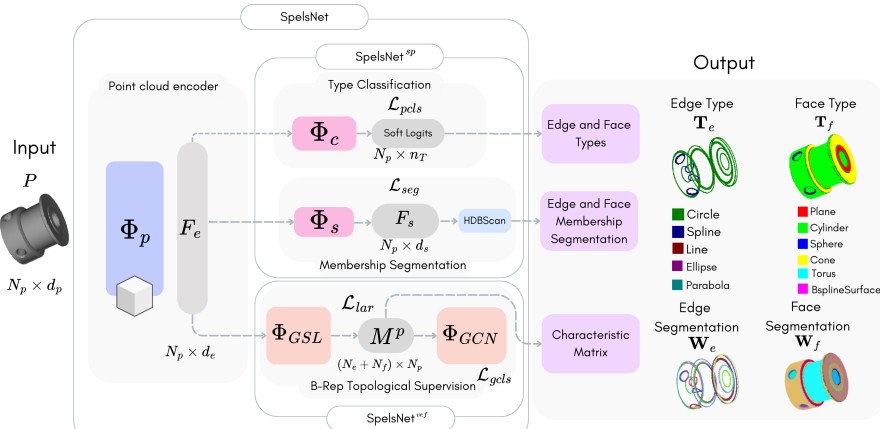

Figure 2: SpelsNet architecture overview. The SparseCNN encoder outputs the point-wise spatial embeddings $\mathbf{F}_e$. Primitive types and membership segmentation learning is done in spatial domain in the SpelsNet$^{sp}$ module together with topological supervision by B-Rep-level elements and structure prediction in Graph Neural Network in the SpelsNet$^{vef}$.

LAR$^{brep}$, the adjacency of edges can be computed as $\mathbf{A}_{ee} = \mathbf{M}_1^p \mathbf{M}_1^{pT}$ and the adjacency of faces as $\mathbf{A}_{ff} = \mathbf{M}_2^p \mathbf{M}_2^{pT}$.

Given a point cloud $\mathbf{P}$, the goal of SpelsNet is to predict the corresponding B-Rep structure including the connectivity and adjacency between the primitives (*i.e.* edges and faces) and their types. In addition to per-point face and edge memberships, SpelsNet leverages the proposed formulation of LAR$^{pcd}$ to guide the adjacency learning via topological supervision. The proposed SpelsNet is described in the next section.

## 4 Proposed Network

We design SpelsNet, a network architecture to segment an input point cloud $\mathbf{P}$ into B-Rep elements. The overall structure of SpelsNet is depicted in Figure 2. SpelsNet operates on point clouds with a SparseCNN encoder Choy et al. [2019] and it is composed of two main components: (1) SpelsNet$^{sp}$ operates in the spatial domain and consists of a type classification head and a membership head; (2) SpelsNet$^{vef}$ leverages the point-to-BRep adjacency supervision to learn the B-Rep topology. In the following, the individual components of SpelsNet are described.

### 4.1 Point Cloud Encoding

The point cloud encoder $\mathbf{\Phi}_p$ is composed of sparse 3D convolutions Choy et al. [2019] in geometric space. In practice, we use a SparseCNN encoder module with a ResUnet backbone, implemented as in Choy et al. [2019]. The input point cloud $\mathbf{P}$ is discretized on a voxel grid with a chosen resolution $\rho$, the input features of dimension $d_p$ are the 3D coordinates of each point and optionally its point normal. As a result, the point cloud is encoded into per-point features $\mathbf{F}_e$ of dimension $d_e = 92$.

### 4.2 Spatial Domain Classification and Segmentation, SpelsNet$^{sp}$

**Type Classification:** The first component of the SpelsNet$^{sp}$ classifies each point as belonging to an edge or a face as well as the type of the primitive $\mathbf{T}_p$. This is achieved by decoding the point embedding $\mathbf{F}_e$ with an MLP $\mathbf{\Phi}_c$ and using the soft logits to classify each point into one of $n_T = 11$ types. The $n_T$ types are composed of 4 curve types, 6 surface types and a class for all possible unknown types. As a result, it is possible to deduce whether a point is an edge or a face point from the predicted type $\tilde{\mathbf{T}}_p$. Point-wise primitive element types are learnt with multi-class cross-entropy loss,

$$\mathcal{L}_{pcls} = \frac{1}{N_p} \sum_{i=0}^{N_p} CE(\tilde{\mathbf{T}}_p[i], \mathbf{T}_p[i]) , \tag{2}$$

where $\mathbf{T}_p, \tilde{\mathbf{T}}_p$ stands for ground-truth and predicted primitive types.

**Membership Segmentation:** In order to segment points as belonging to the same curve or surface patch a metric learning approach is followed. As depicted in Figure 2, the point-wise embeddings $\mathbf{F}_e$ are encoded using an MLP $\mathbf{\Phi}_s$ into features $\mathbf{F}_s \in \mathbb{R}^{N_p \times d_s}$ with $d_s = 128$. The learning of $\mathbf{F}_s$ is conducted using a triplet loss $\mathcal{L}_{seg}$. For a triplet of point-embeddings $\mathbf{f}_s^+, \mathbf{f}_s^-, \mathbf{f}_s^a \in \mathbf{F}_s$ of positive, negative and anchor input, respectively, the triplet loss is given by

$$\mathcal{L}_{seg} = max(||\mathbf{f}_s^a - \mathbf{f}_s^+||_2 - ||\mathbf{f}_s^a - \mathbf{f}_s^-||_2 + m, 0) . \tag{3}$$

The default margin value $m$ is set to $0.05$ and for each sample the number of points is restricted to $8000$ for efficiency reasons. At inference time, the clustering step is done using HDBScan McInnes et al. [2017] to segment the points into edge membership $\tilde{\mathbf{W}}_e$ and face membership $\tilde{\mathbf{W}}_f$ that approximate the ground truth memberships $\mathbf{W}_e$ and $\mathbf{W}_f$, respectively.

### 4.3 B-Rep Topological Supervision, SpelsNet$^{vef}$

The topological supervision includes two main modules, a Graph Structure Learning (GSL) layer and a Graph Convolutional Network (GCN). The GSL aims to learn the point-to-BRep adjacency $\text{LAR}^{pcd}$, whereas the GCN learns the B-Rep element types.

**Graph Structure Learning (GSL):** Inspired by the idea of dynamically learning to construct a graph from a point cloud Wang et al. [2019], we develop a method to connect the spatial 3D shape features, $\mathbf{F}_e$, with the learning of a graph structure that reflects the B-Rep topology. In particular, the goal of the GSL layer is to learn the characteristic matrices of $\text{LAR}^{pcd}$, *i.e.* $\mathbf{M}_1^p$ and $\mathbf{M}_2^p$. In order to facilitate the learning, the matrices $\mathbf{M}_1^p$ and $\mathbf{M}_2^p$ are concatenated in a row-wise manner to form a single matrix $\mathbf{M}^p \in \{\mathbf{0}, \mathbf{1}\}^{(N_e + N_f) \times N_P}$. Given the matrix of per-point point cloud embeddings $\mathbf{F}_e = N_p \times d_e$ where $N_p$ is the number of points in the point cloud and $d_e$ is the dimension of each point embedding, the GSL layer $\mathbf{\Phi}_{GSL}$ predicts the following *weighted* characteristic matrix,

$$\tilde{\mathbf{M}}^p = \mathbf{\Phi}_{GSL}(\mathbf{F}_e) = \texttt{LeakyReLU}(\texttt{Tanh}(\texttt{MLP}(\mathbf{F}_e))) . \tag{4}$$

In the initial experiments, the `ReLU` activation, was utilized to directly enforce sparsity on the output. Due to stability issues discovered during training, this was changed in further experiments to `LeakyReLU`, for which outputs are further clamped to $0$ as minimum value. The use of `Tanh` is advocated by the finding that empirically $\text{LAR}^{pcd}$ with both positive and negative weights gives better results than other options. We employ direct supervision induced by $\text{LAR}^{pcd}$ with an $l1$-loss defined by

$$\mathcal{L}_{lar} = ||\tilde{\mathbf{M}}^p - \mathbf{M}^p||_1 . \tag{5}$$

**Graph Convolutional Network (GCN):** Once the characteristic matrix $\tilde{\mathbf{M}}_p$ has been obtained from the GSL layer, it is possible to leverage these topological features to build an adjacency graph in order to predict the B-Rep elements such as the edge types (*e.g.* lines, spline) and face types (*e.g.* plane, cylinder). As mentioned in Section 3.2, the edge and face adjacency matrices, $\mathbf{A}_{ee}$ and $\mathbf{A}_{ff}$, can be obtained from $\mathbf{M}_1^p$ and $\mathbf{M}_2^p$, respectively. These adjacency matrices are combined into one matrix given by $\tilde{\mathbf{A}} = \tilde{\mathbf{M}}_p \tilde{\mathbf{M}}_p^T$ of dimension $\dim(\tilde{\mathbf{A}}) = (N_e + N_f) \times (N_e + N_f)$. A two-layer GCN $\mathbf{\Phi}_{GCN}$ is introduced to exploit the graph structure inferred by GSL and defined by $(\tilde{\mathbf{M}}_p, \tilde{\mathbf{A}})$. The main idea is to further supervise this graph with an additional head via B-Rep element types. The initial node features of the graph are obtained by a row-wise mean pooling `Pool`$(.)$ of $\tilde{\mathbf{M}}^p$. The graph embedding $\mathbf{Z}$ are learnt according to

$$\mathbf{Z} = \mathbf{\Phi}_{GCN}(\tilde{\mathbf{M}}_p, \tilde{\mathbf{A}}) = \tilde{\mathbf{A}}\texttt{ReLU}(\tilde{\mathbf{A}}\texttt{Pool}(\tilde{\mathbf{M}}_p)\mathbf{\Theta}^0)\mathbf{\Theta}^1 , \tag{6}$$

where $\mathbf{\Theta}^0$ and $\mathbf{\Theta}^1$ are learnable parameters. The embedded $\mathbf{Z}$ is finally passed to a `Softmax` layer with a number of nodes equal to that of primitive types $n_T$. Finally, a primitive type classification cross entropy loss $\mathcal{L}_{gcls}$ is introduced on the output of `Softmax` similarly to $\mathcal{L}_{pcls}$ in Eq. (2).

**Total Loss:** The overall network SpelsNet is trained in an end-to-end manner and the loss is given by,

$$\mathcal{L}_{total} = \alpha_1 \mathcal{L}_{pcls} + \alpha_2 \mathcal{L}_{seg} + \alpha_3 \mathcal{L}_{gcls} + \alpha_4 \mathcal{L}_{lar} . \tag{7}$$

with $\alpha_1$, $\alpha_4$ set to 1 and $\alpha_2 = \alpha_3 = 2$.

# 5 Experiments

## 5.1 Experimental Setup

**ABCParts-VEF Dataset:** SpelsNet is trained and evaluated on the ABCParts dataset Li et al. [2019] using the same train ($22k$), test ($3.5k$) and validation ($3.5k$) splits. We prepare the updated version of this dataset, the ABCParts-VEF dataset, by extending it with B-Rep structural information in the form of characteristic matrices $\mathbf{M}_1^p$ and $\mathbf{M}_2^p$. Refer to the supplementary materials for further details.

**CC3D-VEF Real Scan Dataset:** To evaluate the ability of SpelsNet to generalize to real-world data, a cross-dataset experiment on the proposed CC3D-VEF dataset is conducted. The CC3D Cherenkova et al. [2020] dataset contains 3D scans along with corresponding B-Rep. Similar to ABCParts-VEF dataset, we extend the CC3D dataset with the B-Rep topological information. This proposed version of the dataset is referred to as CC3D-VEF. Testing the model using 3D scans offers an opportunity not only to evaluate how the model generalizes to out-of-distribution data, but also to evaluate how the presence of realistic artifacts such as missing parts, smooth edges, and noise affects the performance. Details of the proposed ABC-VEF and CC3D-VEF datasets are provided in supplementary materials.

**Training and Inference:** The input point cloud is normalized to unit sphere, randomly rotated and discretized on a voxel grid with a resolution $\rho = 0.01$. SpelsNet is trained with AdamW solver with a cosine annealing learning rate schedule starting at $10^{-3}$ and weight decay $10^{-2}$ for 250 epochs to convergence. The training takes approximately 10 days on a node with 4 Nvidia $A100(40Gb)$ GPUs. In order to facilitate the learning, we set the number of edges to $N_e = 128$ and faces to $N_f = 128$. The average inference time per model is $0.5\,$s per model.

## 5.2 Classification and Segmentation Evaluation

In this section, we evaluate the results of SpelsNet on the per-point classification and segmentation tasks against state-of-the-art methods. In this context, only the face type output $\mathbf{T}_p$ and face segment output $\mathbf{W}_f$ from the SpelsNet$^{sp}$ module are considered.

**Baselines:** The results are compared to state-of-the-art methods, namely, ParSeNet Sharma et al. [2020], HPNet Yan et al. [2021] on patches and PrimitiveNet Huang et al. [2021] on surface patches and boundary. For the first two methods we use the checkpoints and datasets, provided by the authors. PrimitiveNet does not provide full training and testing data, thus it was retrained on ABCParts-VEF. We also assess ComplexGen Guo et al. [2022] by obtaining per-point segmentation and type labels from its predictions, transferring them to the original point cloud in a nearest neighbor manner. This enables the alignment of the datasets and metrics, that were not reported in ComplexGen paper due to the implementation differences.

**Test-time Augmentation:** When evaluating Scan-to-Brep in the context of reverse engineering, it is crucial to consider that 3D scans or 3D reconstructions from methods like Multi-View Stereo Seitz et al. [2006] or Nerfs Mildenhall et al. [2021] often lack the alignment to standard axes found in CAD designs. Therefore, in addition to the usual assessment using aligned point clouds (*w/o aug*), evaluating performance under random input rotations (*w/ aug*) is a key indicator of how well the method generalizes to real-world, unaligned data. Typical 3D scanning artifacts(*e.g.* noise, missing parts and details smoothing) are well represented in CC3D dataset Cherenkova et al. [2020].

**Metrics:** We evaluate the per-point classification and segmentation using the same metrics as in Huang et al. [2021], Yan et al. [2021], Sharma et al. [2020]. These include mean type IoU denoted as $tIoU$ and mean segmentation IoU denoted as $sIoU$. More details are in the supplementary.

**Results:** Table 1 summarizes the quantitative evaluation results on the ABCParts-VEF and CC3D-VEF test sets. Clearly, the results demonstrate that all methods have learnt the dataset bias to a different extent. Such, in the presence of unconventional alignment of input point cloud the performance of ParSeNet Sharma et al. [2020] and HPNet Yan et al. [2021] drops significantly. Contrary, PrimitiveNet Huang et al. [2021] and SpelsNet demonstrate more stable results under augmentation by rotation. Our method performs superior in terms of segmentation metrics, and more evidently, in primitive types prediction. Visual results in Figure 3 depict the curves, patches segments along with their type for GT data, and the predictions of SpelsNet , PrimitiveNet and ComplexGen methods.

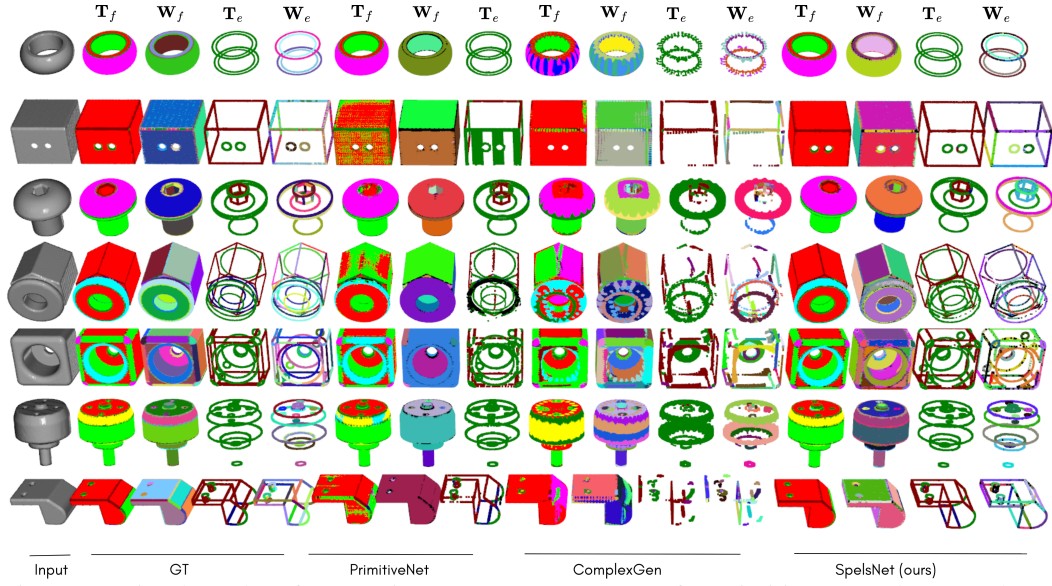

| $\mathbf{T}_f$ | $\mathbf{W}_f$ | $\mathbf{T}_e$ | $\mathbf{W}_e$ | $\mathbf{T}_f$ | $\mathbf{W}_f$ | $\mathbf{T}_e$ | $\mathbf{T}_f$ | $\mathbf{W}_f$ | $\mathbf{T}_e$ | $\mathbf{W}_e$ | $\mathbf{T}_f$ | $\mathbf{W}_f$ | $\mathbf{T}_e$ | $\mathbf{W}_e$ |

Input — GT — PrimitiveNet — ComplexGen — SpelsNet (ours)

Figure 3: Visual results of comparisons on ABCParts-VEF for PrimitiveNet and our SpelsNet. From-left-to-right: input point cloud, face types ($\mathbf{T}_f$) and segmentation ($\mathbf{W}_f$), edge types ($\mathbf{T}_e$) and segmentation ($\mathbf{W}_e$).

## 5.3 Topology Evaluation

In this section, we evaluate the topology predictions of SpelsNet and provide a comparison with ComplexGen Guo et al. [2022] method.

**Baselines:** SpelsNet is the first end-to-end trainable network that predicts the topological connectivity of a B-Rep given an input point-cloud at per-point level. ComplexGen Guo et al. [2022] predicts B-Rep topological elements as well but rather at B-Rep-level. Namely, ComplexGen generates parametric curves and surfaces that correspond to B-Rep elements, along with their topological relationships (vertices, edges, face connectivity) represented by adjacency matrices. Then, topology prediction is compared against ground truth for matched elements, and further topological optimization ensures a valid B-Rep structure. SpelsNet decomposes the input point cloud based on per-point labels obtained from nearest B-Rep elements as the ground truth. It uses B-Rep element connectivity for segmentation supervision at point-level, constructing point-to-B-Rep adjacency. The core idea of SpelsNet is to exploit GCNs to capture relationships between B-Rep elements based on the adjacency reformulation directly within a point cloud data.

**Metrics:** The evaluation of the topological predictions is done using metrics described in Complex-Gen Guo et al. [2022] paper. Specifically, we compute the type accuracy for both edges and faces using the predictions $\tilde{\mathbf{T}}_e$ and $\tilde{\mathbf{T}}_f$ of the GCN. To evaluate the prediction of the topology, we consider the face-edge connectivity in $\text{LAR}^{brep}$ using the characteristic matrix $\mathbf{M}' \in \{\mathbf{0}, \mathbf{1}\}^{N_f \times N_e}$. The

| | ABCParts-VEF | | | | CC3D-VEF | |
|---|---|---|---|---|---|---|
| | Face *w/o aug* | | Face *w/aug* | | Face *w/aug* | |
| Method | sIoU↑ | tIoU↑ | sIoU↑ | tIoU ↑ | sIoU↑ | tIoU↑ |
| ParSeNetSharma et al. [2020] | 78.19 | 81.86 | 34.49 | 45.99 | 13.42 | 18.16 |
| HPNet Yan et al. [2021] | 40.71 | 83.25 | 21.72 | 14.93 | 11.34 | 10.17 |
| PrimitiveNet Huang et al. [2021] | 60.85 | 69.72 | 55.75 | 55.39 | 15.22 | 11.25 |
| ComplexGen Guo et al. [2022] | 33.08 | 45.92 | 32.17 | 45.40 | 14.47 | 19.66 |
| SpelsNet (ours) | 65.72 | 82.35 | 65.60 | 81.93 | 21.23 | 45.15 |

Table 1: Evaluation results on face type and segmentation for the ABCParts-VEF and CC3D-VEF datasets. The metrics are averaged over 5 runs with different random seeds.

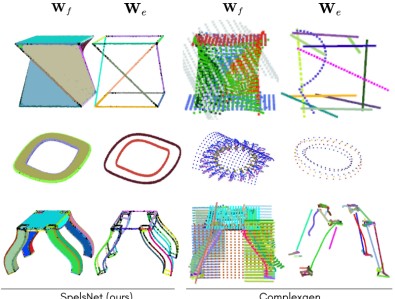

| | Type Acc↑ | | |
|---|---|---|---|
| Method | Edge | Face | $t_{fe} \downarrow$ |
| ComplexGen | 76.3 | 74.2 | 0.201 |
| SpelsNet (ours) | 77.22 | 90.01 | 0.06 |

Figure 4: Qualitative results for face and edge segmentation for ComplexGen and our method.

Table 2: Evaluation of topology results on ABCParts-VEF dataset.

value of $\mathbf{M}'[i, k]$ is 1 if the face $f_i$ is bounded by the edge $e_k$, and 0 otherwise. This characteristic matrix can be easily computed from the predictions of SpelsNet as $\mathbf{M}' = \mathbf{M}_2^p \mathbf{M}_1^{\mathbf{p}^T}$. Note that unlike ComplexGen, we do not need to compute the matching pairs within each group of primitive elements as it is inherently defined by our representation. We define the error $t_{fe}$ of the predicted topological structure $\tilde{\mathbf{M}}'$ with respect to the ground truth matrix $\mathbf{M}'$ as

$$t_{fe} = \frac{1}{N_f N_e} \sum_{i \in N_f, j \in N_e} |\mathbf{M}'(i, j) - \tilde{\mathbf{M}}'(i, j)| . \tag{8}$$

Metrics presented in Table 2 are fully aligned for both our SpelsNet and ComplexGen approaches.

**Results:** The results are reported in Table 2. SpelsNet achieves superior performance in its topology reconstruction as well as type prediction. A visual comparison of predictions for both methods is illustrated in Figure 4. The examples with segmentation results on faces and edges here are obtained from unaugmented results publicly shared by the authors of ComplexGen. The discontinuity artifacts in the B-Rep reflect the higher topological error compared to our method.

## 5.4 Ablation Study

In order to demonstrate the advantage of the joint learning of the SpelsNet$^{sp}$ and SpelsNet$^{vef}$ module, we conduct the following experiment. Three different SpelsNet models are trained: 1) SpelsNet$_p^{sp}$: SpelsNet without the SpelsNet$^{vef}$ module and only point coordinates as input features ($d_p = 3$), 2) SpelsNet$_{pn}^{sp}$: SpelsNet$_p^{sp}$ with added point normals to the input features ($d_p = 6$) and 3) SpelsNet$_{pn}^{sp+vef}$: all the components of SpelsNet with point coordinates and normals as input features. Moreover, the test data is augmented with random rotation (*w/aug*). The results are shown in Table 3. Adding the point normal slightly increases the segmentation results for both edges and types. These segmentation results are further increased by the joint supervision of the two modules of the network, spatial and topological.

**Voxel resolution sensitivity:** The input point cloud, $\mathbf{P}$, is discretized into a voxel grid with a quantization size $\rho$, which determines the size of each voxel in the unit grid. A default value of $\rho = 0.01$ was chosen, balancing model training time and geometric detail resolution on the ABCParts dataset. To investigate resolution sensitivity, the model was evaluated on test data quantized at levels $2\rho$ and $\frac{1}{2}\rho$, with all other settings held unchanged. The results, summarized in Table 4 and Figure 5, indicate the model's sensitivity to the input resolution. Furthermore, we show that the robustness could be enhanced by using a dynamic resolution with respect to adequate selection of voxel density $\psi$ (the average number of points per occupied voxel), during testing. For our backbone, an optimal voxel resolution corresponds to a voxel density $\psi$ of $4 - 6$ points per voxel. This improves testing metrics compared to a fixed resolution, without retraining the model. Future work could explore dynamic resolution selection strategies to further enhance the model's adaptability to varying input data during training.

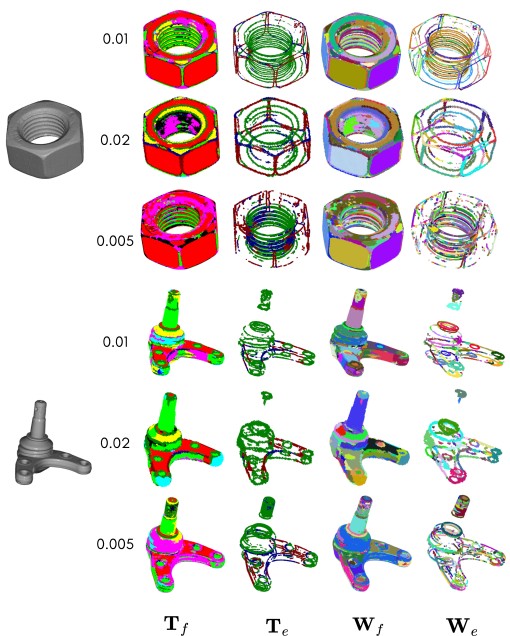

| | | Edge | | Face | |
|---|---|---|---|---|---|
| Method | | sIoU↑ | tIoU↑ | sIoU↑ | tIoU↑ |
| SpelsNet$_p^{sp}$ | | 45.47 | 73.08 | 58.53 | 82.24 |
| SpelsNet$_{pn}^{sp}$ | | 46.69 | 72.86 | 59.69 | 81.26 |
| SpelsNet$_{pn}^{sp+vef}$ | | 50.01 | 72.34 | 65.60 | 81.93 |

Table 3: Ablation studies of SpelsNet on ABCParts-VEF dataset.

| | Edge | | Face | |
|---|---|---|---|---|
| $\rho$ | sIoU↑ | tIoU↑ | sIoU↑ | tIoU↑ |
| 0.02 | 35.60 | 46.56 | 56.59 | 59.04 |
| 0.005 | 43.76 | 61.80 | 61.86 | 61.96 |
| 0.01 | 50.01 | 72.34 | 65.60 | 81.93 |
| $\psi$ | | | | |
| 1-3 | 47.14 | 59.55 | 58.91 | 66.13 |
| 4-6 | 51.54 | 73.22 | 65.45 | 83.74 |
| 7-10 | 45.16 | 49.41 | 59.16 | 72.07 |

Table 4: SpelsNet$^{sp+vef}$ ablation studies results on the ABCParts dataset with respect to voxel quantization size $\rho$ and voxel density $\psi$.

Figure 5: SpelsNet results on real scanned data with respect to various voxel quantization size $\rho$=0.01 (default), 0.02, 0.005.

## 5.5 Discussions and Limitations

The degradation of performance under rotation can be evaluated as a negative outcome. We argue, that uncanonical alignment, specific to real scanned data, offers a way to effectively enlarge the training data and to generalize to unseen data. The CC3D dataset was chosen to demonstrate the model's generalization and robustness to realistic data as it holds a large-scale collection of 3D CAD models and their corresponding 3D scans, exhibiting realistic artifacts like missing parts, surface noise, and smoothed details. The sparse spatial representation allows us to support the input data of dynamic resolutions. While the spatial and topological components of SpelsNet ultimately produce equivalent B-Rep predictions, the topological module was initially introduced for supervising B-Rep elements segmentation. Various other attributes available from B-Rep can be helpful for topological supervision, including sharpness of edges, connectivity degrees, surface area, convexity/concavity of faces etc. In our experiments, the spatial module's predictions outperform the topological module. This could be attributed to insufficient capacity of the GCN network. One of the major limiting factors of our method in terms of learning a highly varied graph-structure is the choice of characteristic matrix of a fixed size, implying that this size should be adjusted according to the data distribution for each new dataset. As future directions, we acknowledge several experiments that could be a part of further SpelsNet performance improvement: (1) Training on CC3D data and validating on other datasets to estimate the effect of different data augmentations and artifacts; (2) The thorough investigation of a spatial backbone choice; (3) The GNN powered by Transformers Kim et al. [2022] is a promising direction to enhance the topological supervision part.

## 6 Conclusions

We present a novel learning approach to B-Rep elements segmentation and type prediction from point cloud data. Our design incorporates features of traditional 3D spatial learning with direct topology supervision through a Graph Neural Network. This is achieved by extending the point cloud data with its corresponding B-Rep structure, using an efficient reformulation based on Linear Algebraic Representations. This unified representation allows us to combine spatial convolutional and graph convolutional networks in a single end-to-end trainable architecture. This leads to more accurate and structurally consistent reconstruction of CAD models from point cloud data and realistic 3D scans.

## Acknowledgement

The present project is supported by Artec3D, and the National Research Fund, Luxembourg under the BRIDGES2021/IS/16849599/FREE-3D, IF/17052459/CASCADES.

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
