# OpenReview forum: "SpelsNet: Surface Primitive Elements Segmentation by B-Rep Graph Structure Supervision"
_NeurIPS.cc/2024/Conference — NeurIPS 2024 poster_

### Official Review · Reviewer_9eMm · 2024-07-09

**Soundness:** 3
**Presentation:** 3
**Contribution:** 3
**Rating:** 6
**Confidence:** 4

**Summary:**

This paper proposes SpelsNet, a novel point-to-BRep adjacency representation learning method that integrates conventional Linear Algebraic Representation of B-Rep graph structures into the point cloud domain. SpelsNet consists of two main components: a supervised 3D spatial segmentation head that classifies B-Rep element types and memberships, and a graph-based head that utilizes the proposed topological supervision. To facilitate SpelsNet's learning, the paper extends two existing CAD datasets with necessary annotations and conducts extensive experiments. The results demonstrate SpelsNet's effectiveness and superiority in achieving accurate and topologically consistent segmentation results compared with existing baselines and state-of-the-art methods.

**Strengths:**

1. This paper uses the boundary information by a Linear Algebraic Representation and combines it into a neural network framework, which is novel and has clear explanations.

2. To learn more discrmitive and informative features, this paper leverages GNN to learn from the edge and face matrix. This sparse matrix is exactly an graph, so it can be greatly handled by GNN.

3. Compared with existing baselines, this proposed SpelsNet achieves impressive improvements in most cases.

**Weaknesses:**

1. The paper only cosiders CAD models, which have nearly no noises, occlusions and distortions. Although very good performance has been achieved, but there is a lack of validation on more realitsitc data.

2. Some parts are confusing and need to be clarified:

2.1. In Eq.(4), the input is point feature, but the prediction of the connectivity matrix should consider multiple points, there is not connectivity on one point.

2.2. In Eq.(4), the paper uses leaky relu for sparsity, but leaky relu cannot output sparse results. They should use relu here.

2.3. The segmentation loss for predicting primitive types has been used more than once in this framework, like in pcls loss and gcls loss, and seg loss also has similar effects. So it’s not clear which predictions should be taken in inference.

2.4 In Section 5.2, all results come from the SpelsNet sp part, why not use vef part?

3. In line 260, the paper mentions that it costs 10 days on 4 A100 GPUs for training. Since the SparseCNN is not a big model, why does it cost so significant time and computation?

**Questions:**

The paper aims to predict primitives including surfaces and edges. It’s not clear why the edges need to be predicted? In practical applications, the engineers may only need to design and combine surfaces. And a second reason is once surfaces are determined, the edges are also fixed.

**Limitations:**

More realistic scenarios like occlusion, noisy and shape variations should also be considerd.

---

> ### Author Rebuttal · Authors · 2024-08-05
>
> 1. The CC3D-VEF dataset is a large-scale collection of 3D CAD models and their corresponding 3D scans. Many realistic artifacts, such as missing data, surface noise, smoothing of sharp details are intrinsic to it (see Figure 3 in supplementary material). The CC3D-VEF dataset was considered with exactly this incentive - to demonstrate the robustness to more realistic data. The cross-dataset experimental results on the CC3D-VEF dataset can be found in Table 1 of the main paper, in which it can be observed that SpelsNet is more robust to such realistic artifacts on the input test data than other SOTA models. The training with this data is left for future work. Additionally, more experiments with varying voxel resolution, another essential aspect of real data analysis, are presented in Table 2 of the rebuttal text (see answer to Reviewer LGxC for more details).
>
> 2.1. $\textbf{F}_e$ is a matrix with per-point embeddings of a point cloud. The dimension of $\textbf{F}_e$ is $N_p \times d_e$ where $N_p$ is the number of points in the point cloud and $d_e$ is the dimension of each point embedding. This will be clarified in the final version of the paper.
>
> 2.2. The remark is totally correct. In the initial experiments, the ReLU activation, was utilized to enforce sparsity on the output. Due to stability issues discovered during training, this was changed in further experiments to LeakyReLU, for which outputs are further clamped to $0$ as minimum value. The statement of sparseness in Eq.4 in its current form does not hold and will be clarified in the final version of the paper.
>
> 2.3. The topological module SpelsNet$^{vef}$ is introduced to supervise the B-Rep elements segmentation of SpelsNet$^{sp}$ and provide relevant signal to the point cloud features $\textbf{F}_e$. As shown in Table 3 of the main paper, the additional supervision by SpelsNet$^{vef}$ leads to an improvement of the results, in particular for the edge and face point segmentation task of the spatial module SpelsNet$^{sp}$. While we chose the supervision of SpelsNet$^{vef}$ to output similar results to SpelsNet$^{sp}$, it is technically possible to conduct it alternative ways. For instance, one can opt to supervise for adjacency alone, or a different criteria considering other attributes available from B-Rep, such as sharpness of edge features, the degrees of connectivity, surface area, convexity/concavity of elements or other attributes. This clarification will be included in the final version of the paper.
>
> 2.4. In our experiments, we found that the predictions of spatial module SpelsNet$^{sp}$ are more favorable in term of performance. One explanation is that the capacity of the GCN module is not enough to perform on par with its spatial counterpart. We plan to investigate this further in future work.
>
> 3. There are several factors that impact the overall training time. The model's robustness to variations in input data orientation is achieved by incorporating random rotation augmentations during training. Additionally, noise injection techniques are used to enhance generalization to more realistic CC3D-VEF data. These augmentations make the convergence slower.  Another contributing factor to the extended training time is the high-resolution nature of the input data, which contrasts with methods designed for segmenting point clouds with significantly fewer points ($2k$-$8k$). In this case, we are working with an average of approximately $50k$-$100k$ points per point cloud. Finally, the dynamic batch size inherent to sparse convolution frameworks presents a challenge in optimizing computational resources, as sufficient memory must be allocated to accommodate peak usage.
>
>
> In CAD applications, B-Rep models can be constructed as the combination of surfaces, where the edges are calculated as a result of their intersections. The final geometry is commonly described via a Boundary Representation. The overall B-Rep structure is comprised of connected elements, i.e., vertices $V$, edges $E$ and faces $F$. The Scan-to-BRep problem can be approached from various directions. One line of work is to recover the surface patches alone. It is followed by state-of-the-art methods such as ParSeNet. An alternative line of work could be described as the detection of edge elements solely. Then the surface patches are defined accordingly as the surfaces that are bounded by the closed loops within the set of edges (e.g. PrimitiveNet). Either way, reconstructing the B-Rep from a point cloud has to address the issue of disjoint surfaces and edges that is a limitation of the aforementioned works. One of the main objective of SpelsNet is to compensate for disjoint artifacts on the reconstructed elements by supervising their connectivity. We argue, that incorporating both edges and faces ensures for more accurate reconstruction of B-Rep structure.

---

> > ### Comment · Reviewer_9eMm · 2024-08-12
> >
> > Thanks for the rebuttal. I would keep my positive rating.

---

### Official Review · Reviewer_PwTs · 2024-07-13

**Soundness:** 3
**Presentation:** 3
**Contribution:** 3
**Rating:** 5
**Confidence:** 4

**Summary:**

This paper proposes a method to segment and classify surface primitive elements (B-rep) from point cloud data. Previous approaches independently deal with the surface patches or boundary curves and ignores the full B-rep structure. This leads to inaccurate and disjoint primitive approximation of the surface. Unlike these approaches, this paper proposes a method to consider topological information of the 3D model through direct supervision during the B-rep learning process. For this, the paper adapts the Linear Algebraic representation of B-rep chain complex into a point-to-B-rep adjacency representation. A thorough experimental validation on ABCparts and CC3D datasets is shown showcasing the improvements over the current methods. Additionally, this method yields compelling results in scenarios where 3D models are not aligned with standard axes, a common occurrence in CAD models, thus proving its efficacy for real-world scans.

**Strengths:**

1. **Clarity:** The paper is well written with each component of the method explained clearly. In addition to this, B-rep representation is adequately explained along with the proposed point-to-B-rep representation in Section 3.2.
2. **Quantitative Analysis:** The approach has been validated against variety of contemporary approaches. Results are comparable with previous methods like ParSeNet [1] and HPNet[2] when the 3D model is aligned with its axes. However, when the data is augmented with transformation SpelsNet achieves SOTA performance which is fair.


[1] Sharma, Gopal, et al. "Parsenet: A parametric surface fitting network for 3d point clouds." ECCV, 2020. \
[2] Yan, Siming, et al. "Hpnet: Deep primitive segmentation using hybrid representations." ICCV, 2021.

**Weaknesses:**

1. **Lack of clarity on ground truth:** The information on how exactly the ground truth is obtained to train the model is not clearly explained. For e.g. it would be good if the paper had some explanation on how the point-to-Brep ground truth is obtained for supervision in Eq. 5 for $L_{lar}$ loss.
2. **Quantitative Evaluation:** Although quantitative results for face type and segmentation are provided for the previous methods. Quantitative comparison with ComplexGen (the most recent one) is missing in Table 1.
3. **Intuition behind the generalization and robustness:** The model shows strong generalization ability as results are shown when the model is trained on ABC dataset and tested on CC3D and real scans (Table 1 in main paper and Fig 3 in supplementary). In addition to this the model shows robustness especially in cases where the 3D model is not aligned with the axes. However, it is not very clear from the paper how this generalization and robustness is achieved. A brief information on this will be very helpful.
4. **Qualitative Evaluation:** Qualitative results in Fig. 3 is shown only PrimitiveNet and SpelsNet. Additional results of other methods will be helpful.

**Questions:**

1. **Training time:** Although not a big problem, was curious about the training time of the model. Despite training on only 22k models, it takes 10 days on 4 Nvidia A100 40 Gb gpus. What is the reason behind this long training time?


Please follow the weakness for more points.

**Limitations:**

The paper discusses the limitations adequately.

---

> ### Author Rebuttal · Authors · 2024-08-05
>
> **Ground Truth Preparation**: Details on data preparation were included in Section 3.1 of the supplementary material due to space constraints. The labeling information of a B-Rep structure is transferred into its mesh representation using a per triangle nearest neighbor assignment under the tolerance threshold $\tau = 0.008$. The B-Rep level topology in the form of $LAR^{brep}$ is extracted directly from B-Rep $V$, $E$, $F$ elements relation. The point-to-BRep adjacencies are calculated as described in Section 3.2 of the main paper, where $LAR^{pcd}$ is represented by its two characteristic matrices $\mathbf{M}_1^p$ and $\mathbf{M}_2^p$. The matrices $\mathbf{M}_1^p$ and $\mathbf{M}_2^p$ are row-wise concatenated into a single matrix for the supervision in Eq.5.
>
> **Quantitative and Qualitative evaluation**: For a fair comparison with ComplexGen, we need to note first that the implementation differences between ComplexGen and SpelsNet do not allow to account the same $sIoU$ and $tIoU$ metrics as for the methods reported in Table 1 of the main paper. More clarification on the main differences in the predictions of SpelsNet and ComplexGen can be found in the reply to the comments of Reviewer js6S. To compute the same metrics on the same data, we have to obtain the per-point segmentation and type labels from the predictions of ComplexGen. Similarly to data preparation, we compute this by transferring predictions from the approximated curves and patches in a nearest neighbor manner into the original input point cloud. The quantitative results are supplemented in Table 1 of this rebuttal. The visual results are also amended to include the ComplexGen predictions obtained (Figure 1 of this rebuttal text). Those metrics were not presented in the original ComplexGen work for the reasons mentioned above. Thus, we initially opted to compare with this work on the metrics (Table 2 of the main paper) that are strictly aligned with our approach. We will clarify this point in the final version of the paper.
>
> **Generalization and Robustness**: The robustness towards input data orientation is achieved by training with random rotation augmentations. Additionally, noise injection allow for better generalization to more realistic CC3D-VEF data. We also point out that even better results could be obtained when training on the CC3D-VEF dataset (Section 5.5 of the main paper).
>
> **Training time**: The convergence of the model during training when rotation and noise augmentations are added is significantly slower than without augmentations. Another reason for a significantly large training time is the high resolution nature of the input data compared to the methods that learn to segment point cloud with $2k$-$8k$ points. On average, the input point cloud size is $50k$-$~100k$ points. The last factor in the slow convergence is the dynamic nature of the batch size that is dealt in sparse convolution framework. It typically does not allow to utilize computation resources optimally, as we have to guarantee for an amount of memory available at peak usage. For comparison, ComplexGen takes 3 days on 8 Nvidia V100 GPUs to converge in 500 epochs.

---

> > ### Comment · Reviewer_PwTs · 2024-08-12
> >
> > Thanks for the rebuttal. After reading the rebuttal and other comments by the reviewers, I will stick to my original score!

---

### Official Review · Reviewer_js6S · 2024-07-13

**Soundness:** 2
**Presentation:** 2
**Contribution:** 3
**Rating:** 6
**Confidence:** 4

**Summary:**

This paper focused on reverse engineering where cad brep is reconstructed from point cloud. Authors extended the definition of face-edge-vertex incidence matrix to point-face, point-edge adjacency. A new SpelsNet module is added on top of existing pipeline to also predict point assignment w.r.t the primitives. They showed that the predicted topology can be passed to another gcn to better predict the brep components.

**Strengths:**

Paper is well-written and easy to follow. The extension from brep adjacency graph to point assignment is straight forward. It is interesting to see that the predicted topology from per-point prediction can help downstream brep component classification. Results are good versus baseline without the topology prediction module.

**Weaknesses:**

The idea of using predicted topology to help with brep reconstruction has already been proposed by previous method ComplexGen. SpelsNet seems similar in terms of predicting the adjacency matrix. It also has similar results on edges but much better for faces. This is very interesting as the much larger surfaces are usually easier to reconstruct than curves.  The results reported here seem to contradict this. I wonder what would happen is ComplexGen is trained with the same data? Would the difference be much smaller?

**Questions:**

Can authors explain the major difference between SpelsNet and ComplexGen? Both network predicts the topology and show that this is better for the final reconstruction.

How are the comparisons between SplesNet and ComplexGen conducted? Is ComplexGen evaluated using the pretrained public model without any finetuning?

**Limitations:**

Yes, the authors adequately addressed the limitations.

---

> ### Author Rebuttal · Authors · 2024-08-05
>
> Firstly, we would like to clarify the difference between SpelsNet and ComplexGen. Both SpelsNet and ComplexGen demonstrate the importance of topology for B-Rep reconstruction, but they differ in their approach.
>
> ComplexGen generates the B-Rep level geometric primitives as parametric curves and surfaces  along with the topological relations of B-Rep elements, such as vertices $V$, edges $E$ and faces $F$ connectivity. The curve and surface parameters are estimated by fitting the primitive of respective type into the detected points. Single feature vector corresponds to each predicted element within the $V$, $E$, $F$ sets. The connectivity is represented via the adjacency matrices of respective elements, and explicitly constructed as their pairwise dot product. Topology prediction is compared with the ground-truth adjacency topology for the matched elements only. Furthermore, the topological optimization solves for the topological objective and geometric fitness to ensure the structural validity of the final B-Rep chain complex with respect to the explicit topological constraints (see Eq. (1)-(3) in ComplexGen paper). This valid structure is further used in an iterative geometry refinement process where the parameters of the primitives are improved with respect to input point cloud geometry.
>
> SpelsNet follows a different approach. The input point cloud is decomposed into B-Rep elements based on per-point labels assignment. SpelsNet uses B-Rep element's connectivity in the form of two characteristic matrices $\mathbf{M}_1^p$ and $\mathbf{M}_2^p$ for edge and face segmentation supervision respectively. As there is no direct way to relate point cloud connectivity with B-Rep topology, we develop a straightforward method to construct point-to-BRep adjacency, described in Section 3.2 of the main paper. This formulation provides a convenient input for learning the graph structure intrinsic to B-Rep, and enables the Graph Convolution Network (GCN) supervision. The core idea of our approach is to use GCNs as they are well-suited for capturing the inherent relationships between faces, edges, and vertices in B-Rep models. Unlike 3D CNNs, which are designed for regular grids, GCNs can operate on irregular graph structures, such as B-Rep and can be trained to directly predict B-Rep-level labels from raw input data in an end to end manner. This approach simplifies the overall pipeline.
>
> A second point we would like to clarify concerns the comparison between SpelsNet and ComplexGen. Both SpelsNet and ComplexGen are trained and tested on the same splits of the ABCParts data obtained from the same B-Rep and point cloud counterparts. The ground truth data is prepared differently with specifics of each method. The visual example of both predictions is given in Figure 4 of the main paper. SpelsNet predicts per-point type and segment labels along with point-to-BRep adjacency matrices, whereas ComplexGen predicts parametrized curves and surface patches with their respective types, and B-Rep level adjacency matrices. For SpelsNet the type of the predicted segment is defined by a majority voting of the points assigned to it.
>
> For ComplexGen, the ground truth and predictions of B-Rep elements are approximated with a fixed number of points (30 for any curve, and 10x10 for a patch) irrespective of the actual size of the element. While one would expect large surfaces to be easier to reconstruct than edges, the aforementioned design choice in ComplexGen leads to a comparable performance in type accuracy for edges and faces (Table 2 of the main paper). On the other hand, as SpelsNet predicts per-point labels on the input point cloud it achieves a higher face type accuracy than edge type accuracy as expected.
>
> The metrics in Table 2 of the main paper for ComplexGen are evaluated with their publicly shared code using the pretrained model and preprocessed dataset shared by the authors. The metrics for SpelsNet are computed from the outputs of our topological supervision module SpelsNet$^{vef}$.
>
> In Table 1 of the attached document, we also report the performance of ComplexGen with respect to $sIoU$ and $tIoU$ metrics common to other state-of-the-art approaches. The predictions of ComplexGen in this case were transferred into the original input point cloud using a nearest neighbor assignment strategy. The differences between ComplexGen and SpelsNet as well as the clarification on the comparison method will be added to the final version of the paper.

---

> > ### Comment · Reviewer_js6S · 2024-08-12
> >
> > I want to thank authors for the detailed rebuttal. Most of my concerns are resolved. The difference between complexgen and splesnet appear to be per-pritimive VS per-point , with the later approach showing much better results. Overall the method does seem to be a lot simplier than the very "complex" complexgen. I think this is a good improvement and adjusted my score accordingly.

---

### Official Review · Reviewer_LGxC · 2024-07-20

**Soundness:** 3
**Presentation:** 3
**Contribution:** 3
**Rating:** 6
**Confidence:** 3

**Summary:**

This paper presents a novel model called SpelsNet for surface elements segmentation task based on boundary representation, which uses two heads for predicting B-Rep element types and extracting topological information, respectively. Experiments are evaluated on two extended datasets and the results illustrate the efficacy of the proposed method.

**Strengths:**

1.	This work proposes an approach to exploit the topological feature within B-Rep graph structure.
2.	A LAR-based point-to-BRep adjacency supervision is integrated for B-Rep element segmentation supervision.

**Weaknesses:**

1.	The concrete voxel grid resolution for the input seems unclear. How does the performance vary with respect to different resolution settings? A related sensitivity analysis is desired.
2.	The resolution of Figure 2 is a bit low. It’s difficult to see the subscript of notations clearly even already zoomed in.
3.	The topological information supervision based on B-Rep chain complex sounds not a unique way. Methods related to topological optimization should also provide a workaround for this task. Further comparisons and analysis with this approach should be appended.

**Questions:**

Can the proposed method be applied on pure point cloud (without edge connectivity) as input?

**Limitations:**

Detailed discussion about the ablation study is insufficient. The insights behind the proposed modules of SpelsNet should also be discussed here.

---

> ### Author Rebuttal · Authors · 2024-08-05
>
> 1. The input point cloud $\mathbf{P}$ is discretized on a voxel grid with a fixed voxel quantization size $\rho$. The default value is set to $\rho=0.01$ corresponding to the voxel grid size of $100^3$. This value is chosen considering the trade-off between the model's training time and its ability to resolve the geometric details on ABCParts dataset. In order to further investigate the sensitivity of the model to the voxel resolution, we evaluate the SpelsNet$^{sp+vef}$ model on test data quantized with a voxel quantization size $2\rho$ and $\frac{1}{2}\rho$. All other settings remain unchanged. The resulting metrics are summarized in the top section of Table 2 of the attached document. The model is sensitive to the input voxel resolution. In fact, it can be observed that a $\rho$ value of $0.02$ leads to degradation of the results. This resolution is likely too low to capture the fine grained details. The higher resolution ($\rho=0.005$) induces arbitrarily more noise in the predictions. This could be caused by the difference in voxel quantization size between training and testing. In Figure 2 of the attached document the results on real scanned data (no ground truth is known) are presented. Given various voxel quantization size, the proposed method tends to either obliviate intricate details (the thread part of the top model) or to generate noisier segments.
>     The robustness of SpelsNet is further investigated with a dynamic voxel resolution selection. Let $\psi$ be the voxel density, that is the the average number of points belonging to the occupied voxel after quantization. The performance of SpelsNet for different values of $\psi$ at test time can be found in the lower part of Table 2. We find that a voxel density $\psi$ of $4-6$ points per voxel improves the metrics without having to retrain the model compared to a fixed voxel quantization size. These results will be included in the final version of the paper.
>
> 2. The resolution of Figure 2 will be improved.
>
> 3. Topological optimization can explicitly reward or penalize specific topological characteristics. However, topological optimization based methods tend to be computationally intensive, especially for feature-rich models. As an example, one can perform a topology optimization step with explicit constraints inherited from B-Rep graph structure validness (see equations 1-3 in ComplexGen). The topology information generated by ComplexGen in the form of B-Rep chain complex elements is used to guide a topology optimization process. As reported in ComplexGen, this process is computationally intensive, especially for the models with a large number of elements. On average it takes 8-10 minutes for one model to be solved. As a result, topology optimization based methods may have limited practical applications with realistic and complex datasets such as the CC3D-VEF dataset. We opted for an end-to-end neural-based approach to supervise the topology. The computational impact is significantly lower at inference time which makes it more applicable for realistic scenarios.
>
> 4. The proposed method operates on a point cloud without any additional input other than point coordinates (and optional point normals) during inference. The connectivity information is only used during training.

---

### Author Rebuttal · Authors · 2024-08-06

First of all, we express our gratitude to the reviewers for their valuable feedback. We retain that our model SpelsNet is "novel"(LGxC, 9eMm). "The results illustrate the efficacy of the proposed method"(LGxC), and "are good versus baseline without the topology prediction module"(js6S). "The approach has been validated against variety of contemporary approaches"(PwTs) and "compared with existing baselines, ... SpelsNet achieves impressive improvements"(9eMm). The "paper is well-written and easy to follow""(js6S) "with each component of the method explained clearly"(PwTs).

Further, we clarify the specific points in their comments and provide, where possible, the supporting experiments.

**Difference between SpelsNet and ComplexGen**: Overall, SpelsNet proposes a novel approach to exploit the topological information from Boundary Representation (B-Rep) models as a direct neural supervision within a Graph Convolutional Network (GCN) framework for segmenting B-Rep elements (edges and faces) on a point cloud. The approach offers an adaptation of the Linear Algebraic Representation (LAR) of B-Rep chain complex to point clouds via point-to-BRep adjacency formulation, that enables direct supervision of B-Rep topology on point clouds.

Both SpelsNet and ComplexGen leverage topology for B-Rep reconstruction, but with distinct methodologies. ComplexGen generates parametric curves and surfaces that correspond to B-Rep elements, along with their topological relationships represented by adjacency matrices. Topology prediction is compared against ground truth for matched elements, and further topological optimization ensures a valid B-Rep structure. SpelsNet decomposes the input point cloud based on per-point labels obtained from nearest B-Rep elements as the ground truth. It uses B-Rep element connectivity for segmentation supervision at point-level, constructing point-to-B-Rep adjacency. The core idea is to further exploit GCNs to capture relationships between B-Rep elements based on this B-Rep adjacency reformulation directly within a point cloud data. The topological module, SpelsNet$^{vef}$ provides additional supervision that improves the results.

**Additional ComplexGen evaluation**: The differences between ComplexGen and SpelsNet, which prevent a direct comparison using the same $sIoU$ and $tIoU$ metrics, are important. For a fair assessment, we obtained per-point segmentation and type labels from ComplexGen predictions, transferring them to the original point cloud using a nearest neighbor approach. This allows us to compute the same metrics on the same data, with results presented in Table 1 of this rebuttal. We updated the visuals to incorporate these ComplexGen predictions. Notably, the metrics were not presented in the original ComplexGen work due to the aforementioned implementation differences. Thus, our initial comparison focuses on metrics (Table 2 of main paper) that are aligned with our approach.

**Voxel resolution sensitivity**: The input point cloud, $\mathbf{P}$, is discretized into a voxel grid with a quantization size $\rho$. A default value of $\rho=0.01$ was chosen, balancing model training time and geometric detail resolution on the ABCParts dataset. To investigate resolution sensitivity, the model was evaluated on test data quantized at levels $2\rho$ and $\frac{1}{2}\rho$, with all other settings held unchanged. The results, summarized in Table 2 of the rebuttal, indicate the model's sensitivity to the input resolution. Furthermore, we show that the robustness could be enhanced by using a dynamic resolution with respect to adequate selection of voxel density, $\psi$ (the average number of points per occupied voxel), during testing. For the ResNet34 backbone, an optimal voxel resolution corresponds to a voxel density $\psi$ of $4-6$ points per voxel. This improves testing metrics compared to a fixed resolution, without retraining the model. Future work could explore dynamic resolution selection strategies to further enhance the model's adaptability to varying input data during training as well.

**Generalization and robustness**: The CC3D-VEF dataset offers a large-scale collection of 3D CAD models and their corresponding 3D scans, exhibiting realistic artifacts like missing data, surface noise, and smoothed details (Figure 3 in supplementary material). This dataset was specifically chosen to demonstrate the model's robustness on more realistic data. The model's robustness to input data orientation is achieved through training with random rotation augmentations. Furthermore, noise injection improves generalization to more realistic CC3D-VEF data. While training on CC3D-VEF is planned for future work, additional experiments varying voxel quantization size $\rho$ – another crucial aspect of real data analysis – are presented in Table 2 of this rebuttal.

**Training time**: While training with these augmentations improves robustness, it significantly slows down the convergence process compared to training without them. The high resolution of the input data (~100k point clouds on average) further contributes to the extended training time, especially compared to methods designed for 2k-8k point clouds. Additionally, the dynamic batch size inherent in the sparse convolution framework prevents optimal utilization of computational resources due to memory constraints.

**Topological supervision**: While the spatial and topological components of SpelsNet ultimately produce equivalent B-Rep predictions, the topological module was initially introduced for supervising B-Rep element segmentation. Various other attributes available from B-Rep can be helpful for topological supervision, including sharpness of edges, connectivity degrees, surface area, convexity/concavity of faces etc. In our experiments, the spatial module's predictions outperform the topological module. This could be attributed to insufficient capacity of the GCN network, which we plan to investigate in future work.

---

### Decision · Program_Chairs · 2024-09-25

**Decision:**

Accept (poster)

**Comment:**

The paper received positive reviews, with one *borderline accept* and three *weak accept* ratings. The reviewers maintained their positive perspectives after the rebuttal and did not raise any further concerns. Additionally, no ethical issues were identified by the reviewers.

Therefore, the area chair recommends accepting this paper. The authors are encouraged to incorporate the clarifications provided in the rebuttal into the final version to address points such as comparisons with ComplexGen, generalization, voxel resolution sensitivity, and training time.